# An EMR-Based Approach to Determine Frequency, Prescribing Pattern, and Characteristics of Patients Receiving Drugs with Pharmacogenomic Guidelines

**DOI:** 10.3390/pharmacy11060178

**Published:** 2023-11-17

**Authors:** George E. MacKinnon, Megan Mills, Alexander Stoddard, Raul A. Urrutia, Ulrich Broeckel

**Affiliations:** 1School of Pharmacy, Medical College of Wisconsin, Milwaukee, WI 53226, USA; mmills@mcw.edu (M.M.); ubroeckel@mcw.edu (U.B.); 2Clinical & Translational Science Institute, Medical College of Wisconsin, Milwaukee, WI 53226, USA; astoddard@mcw.edu; 3Linda T. and John A. Mellowes Center for Genomic Sciences and Precision Medicine, Medical College of Wisconsin, Milwaukee, WI 53226, USA; rurrutia@mcw.edu; 4Department of Pediatrics, Medical College of Wisconsin, Milwaukee, WI 53226, USA; 5RPRD Diagnostics LLC, Milwaukee, WI 53226, USA

**Keywords:** pharmacogenomic drugs, pharmacogenomic testing, electronic medical records, evidence-based guidelines, physicians, pharmacists

## Abstract

(1) Background: This retrospective analysis utilizing electronic medical record (EMR) data from a tertiary integrated health system sought to identify patients and prescribers who would benefit from pharmacogenomic (PGx) testing based on Clinical Pharmacogenetics Implementation Consortium (CPIC) guidelines. (2) Methods: EMR data from a clinical research data warehouse were analyzed from 845,518 patients that had an encounter between 2015 and 2019 at an academic medical center. Data were collected for 42 commercially available drugs with 52 evidence-based PGx guidelines from CPIC. Provider data were obtained through the EMR linked by specialty via national provider identification (NPI) number. (3) Results: A total of 845,518 patients had an encounter in the extraction period with 590,526 medication orders processed. A total of 335,849 (56.9%) patients had medication orders represented by CPIC drugs prescribed by 2803 providers, representing 239 distinct medications. (4) Conclusions: The results from this study show that over half of patients were prescribed a CPIC actionable medication from a variety of prescriber specialties. Understanding the magnitude of patients that may benefit from PGx testing, will enable the development of preemptive testing processes, physician support strategies, and pharmacist workflows to optimize outcomes should a PGx service be implemented.

## 1. Introduction

Pharmacogenetics (PGx) is the study of how different individual’s genetic polymorphisms can result in alterations in medication response [1,2,3]. The goal of pharmacogenetics is to help prescribers select medications and doses for a patient that are the most efficacious and reduce adverse side effects [4].

In the US, 18% of all prescriptions are influenced by actionable PGx genes, with 90–99% of the population having at least one high-risk variant for established genes [3,5,6,7]. Common reasons for conducting PGx services include toxicity concerns, side effects, medication non-responders (e.g., the lack of efficacy), a family history of significant variants, preemptive to treatment, or incidental research findings [8]. These concerns led to the U.S. Food and Drug Administration (FDA) to include pharmacogenomic labeling information for over 200 medications that have a particular gene–drug interaction, including some that warn of potential life-threatening situations via “Black Box warnings” [9,10]. The FDA has required manufacturer labeling for such known gene–drug interactions that may pose serious risk to patients. Some examples of these commonly prescribed medications include warfarin, ondansetron, and codeine. These medication labels provide information if testing is recommended when prescribing a medication and if there is actionable pharmacogenetics for the medication [11].

The Clinical Pharmacogenetics Implementation Consortium (CPIC) is an international organization made up of scientific and clinical professionals with the goal of promoting the use of pharmacogenetic testing in patient care settings [5]. CPIC has created pharmacogenetic evidence-based guidelines for patient care providers to create standardization in prescribing and dosing in patients with a variety of genetic variants based on published scientific reports. These guidelines focus on the most common and well-known pharmacogenetic associations and aim to be implemented in a variety of practice settings. Patient-specific diplotypes for each gene are categorized into drug-dosing groups and clinical phenotypes [5]. The CPIC has four gradations of evidence (A-D) reported, with “A” representing that genetic information should be used to change the prescription of the affected drug; thus, the preponderance of evidence is high or moderate.

Clinicians can utilize the CPIC to aid in translating genetic laboratory test results into actionable prescribing decisions for established gene–drug pairs, such as suggesting treatment alternatives, as well as providing direction for safe prescribing practices [3,5]. As of 2023, there were 26 published pharmacogenomic-based CPIC guidelines covering 23 genes and 145 drugs across a variety of therapeutic categories [5]. Another resource, funded by the National Institutes of Health (NIH) is the Pharmacogenomics Knowledge Base (PharmGKB), an online resource providing information about variations within human genetics impacting patients’ responses to medications [12].

Pharmacogenetic testing can be performed preemptively and reactively. The majority of PGx testing performed in the US occurs after a patient experiences an adverse effect or has a history of non-response to drug therapy, which is referred to as reactive PGx testing [13]. Often a reactive test will assess only one gene–drug pair; thus, the future utility of the results is limited in accessibility to other prescribers. Contrarily, preemptive PGx testing is proactive, performed to help avoid adverse medication side effects and improve medication efficacy before a specific medication is selected and taken by the patient [6,14,15]. Preemptive testing provides information that may be useful later, as the health conditions and medication needs of the patient change, yet their genotype will most likely remain the same [15,16]. Preemptive pharmacogenetic testing is beginning to be utilized in a number of patient diagnoses, such as oncology, cardiology, hematology, as well as patient populations (e.g., pediatrics). As PGx testing continues to decrease in cost, preemptive testing may be a more cost-effective approach to patient care [3,6,13]. For example, conducting PGx testing in a child would have great utility as the child matures into an adult, thus providing metabolic drug information over their lifetime.

In healthcare settings, it is critical that when such genetic information (e.g., PGx results) is obtained, it is made available to all providers engaged in a patients care and linked with clinical data to guide providers in selecting safe and effective medication therapies [2,3,7]. Thus, having a repository of searchable discrete data including PGx results in an electronic medical record (EMR) is optimal for patient care. Understanding the full magnitude of potential PGx testing could have on a healthcare system, providers and patients is warranted. This retrospective analysis utilizing EMR data from a large tertiary integrated health system sought to identify patients and prescribers who would benefit most from comprehensive PGx testing based on established guidelines (e.g., CPIC). Having conducted an internal analysis would assist decision makers in determining the overall scope for a PGx service if implemented at the medical center should it someday be offered to patients and providers.

## 2. Materials and Methods

EMR data for the academic medical center were de-identified and integrated in the CTSI Clinical Data Warehouse. Prescribing records and demographics were characterized for 845,518 patients with an encounter between 2015 and 2019. The formulary was linked via RxNorm at the ingredient level to find all orders for 42 drugs with 52 PGx guidelines (known at the time of this analysis) from the CPIC, termed PGx drugs. Prescriber specialization was linked using national provider identification (NPI) numbers.

Only ‘new’ patients were characterized to assess the frequency and complexity of PGx ordering for patients who would have been hypothetically naïve to PGx testing at index in this retrospective cohort. A ‘new’ patient was defined to require no history of any PGx drug and an encounter at least one year prior to the first PGx drug order.

Prescribing complexity was characterized by provider specialty, the number of orders, PGx guidelines, ordering frequency, and the multiplicity of PGx drugs. The frequency and duration of ordering for individual drugs were characterized using a heuristic of ‘ordering episode’. All orders of a given drug were grouped into a single episode if each subsequent order occurred within a dynamically calculated duration of 1.5 years from each prior order. This grouped orders recurring annually or more frequently into one episode. Computational date math is exact, while human calendar scheduling is not. Thus, a conservatively selected 0.5-year buffer was chosen ad hoc before any analysis to account for this variation. Multiplicity of ordering of different drugs was characterized using the interval of days between the first order of the first PGx drug and the earliest order of the last PGx drug for individuals with combinations of PGx drugs ordered within the study period, termed the ‘ordering interval’.

## 3. Results

### 3.1. Comparison of Demographics for Patients with Encounters, Medication Orders, and CPIC Drug Orders 2015–2019

Table 1 summarizes the demographic information of the patients’ charts analyzed from the EMR. Females were more represented (54.9%) than males. A total of 845,518 patients had an encounter between 2015 and 2019, and from these encounters there were 590,526 medication orders processed. A total of 335,849 (56.9%) patients had medication orders represented by the 52 CPIC drugs.

Figure 1 displays the total orders of medication grouped by the CPIC guideline drug compared to the ordering provider specialty. Over 19 medical specialties prescribed CPIC-guided medications, with the highest amount of these medications prescribed from the departments of anesthesiology, oncology, internal medicine, and family practice. The larger the area represented, the more prescriptions of the class of medication. Likewise, the larger the representation of a specialty, the more prescriptions are written by the discipline. As seen, ondansetron was the most frequently prescribed medication by anesthesiology and hematology/oncology.

### 3.2. Comparison of CPIC Drug Ordering Characteristics in 2015–2019

As described in Table 2, CPIC drug-ordering characteristics are grouped based on CPIC guideline with unique patients. Ondansetron was the most frequently prescribed medication (1,302,015 orders in 202,113 patients). This was followed by the opioids (hydrocodone, tramadol, and codeine), NSAIDs (ibuprofen, meloxicam, and celecoxib), PPIs (pantoprazole and omeprazole) and SSRIs (sertraline and escitalopram).

Table 2 shows the median number of days between orders for ondansetron, opioids, and the NSAIDs was one day. This median increased for the other categories of medications, with simvastatin having the greatest number of days between orders with a median of 619 days. One ordering interval is defined as all contiguous orders of a given drug for a patient without a gap of more than 1.5 years. The period of 1.5 years was arbitrary but was picked to be very conservative for treatment courses with infrequent provider/patient interaction. 

Ordering episode is an attempt to distinguish and quantify the difference between acutely and chronically administered drugs. It is a dynamic duration representing an inferred period of continuous care/medication exposure; each episode is presumably covered by one provider/patient relationship, and the orders in an episode are expected to be assessed by one PGx test and review. It prevents orders separated in time from distorting the statistics (e.g., two acute care encounters years apart for a patient would be summarized separately).

Ordering interval is a different concept and describes the multi-drug case without any limitation of episode duration. Ordering interval extends the concept of ‘ordering episode’ to more than one drug (with potentially multiple treated conditions and provider relationships), where a comprehensive and possibly preemptive PGx panel test would have the most application. It characterizes the distribution of times for the first orders of each drug (note that it is not defined for the single drug case).

### 3.3. Combinations of CPIC Drugs Ordered in 2015–2019

As seen in Table 3, between 2015 and 2019, a total of 133,380 patients were prescribed at least one CPIC medication with 47 distinct drug combinations. Further, 84,334 patients were prescribed two CPIC medications, and 51,357 patients were prescribed three CPIC medications. An ordering interval is defined as the number of days between the first order of the first drug and the earliest order of last drug in a combination for each patient. Patients who were prescribed three or fewer CPIC drugs were prescribed this combination in less than a year. Interval days becomes a measure of how lead-time and/or turn-around time of a PGx panel test would impact the timely availability of additional PGx results and hence the applicability of PGx panel testing. 

## 4. Discussion

The results from this study show that over half of patients 335,849 (56.9%) were prescribed a CPIC actionable medication within the academic medical center. Many patients were prescribed multiple CPIC drugs from a variety of provider specialties. We characterized the distribution of first orders for the many patients for whom multiple CPIC drugs were ordered within the study period as ‘ordering interval’.

The median number of days between orders for ondansetron, opioids, and the NSAIDs was one day, demonstrating this combination of medications is commonly prescribed at the medical center for many patients undergoing acute care. In contrast, the median number of days between orders increased for other categories of medications, such as simvastatin having the greatest number of days between orders with a median of 619 days. This large time frame suggests that some patients are on this medication for a protracted time.

Between 2015 and 2019, of the 335,849 patients records reviewed, 133,380 (39.7%) patients were prescribed at least one CPIC medication with 47 distinct drug combinations. Further, 84,334 (25.1%) patients were prescribed two CPIC medications, and 51,357 (15.3%) patients were prescribed three CPIC medications. Many of these patients were prescribed medications from a variety of provider specialties. These data help to inform the magnitude of having systems in place to appropriately manage the results of PGx testing that can have implications on many classes of medications.

It is important to consider which patients would be best suited for PGx testing, reactively or preemptively. Because PGx test results may be used by additional providers over time, increasing the overall utility (i.e., retrievability from the EMR) is essential. The magnitude of the number of medical specialties that prescribed CPIC-guided medications is apparent (e.g., departments of anesthesiology, oncology, internal medicine, and family practice).

While arguably many providers have been indoctrinated into the principles of genetics and pharmacology, this level of understanding is not sufficient to address the implications of a clinical PGx result and interpretation. Navigating and understanding the CPIC pharmacogenetic evidence-based guidelines, PharmGKB resources about variations within human genetics impacting patient’s responses to medications, and FDA labeling requirements is daunting with respect to prescribing. This demonstrates the challenges of trying to educate a physician workforce across a large healthcare system.

The number of medications that will continue to be identified for PGx testing and monitoring is certainly not static. Hence, the need to keep abreast of new prescribing updates will be ever-growing among the medical specialties. Recognizing these challenges, two associations from the disciplines of medicine and pharmacy, the American Medical Association (AMA) and the American Association of Health-Sytem Pharmacists (ASHP), created a six-part pharmacogenomic virtual summit series that was offered in 2021 [17].

The clinical significance of PGx testing stems from using evidence-based guidelines, such as the CPIC, and linking patient-specific genotypes to phenotypes and relating those phenotypes to proper medication selection, avoidance, or dose adjustment [2,7]. PGx-guided prescribing can improve the safety of associated outcomes with medication use, and, in some cases, reduce the total cost of care. Improving access to PGx data within the EMR is an essential element to allowing the clinical implementation of PGx into practice [13].

The proposed idea of *right drug, right dose, right patient,* is only possible with the right information, available at the right time as well [9,18]. Clinical decision support tools (CDST) within the EMR are critical tools for the integration of pharmacogenomics into routine patient care [15,19]. The incorporation of CDSTs into the EMR to flag issues or “fire” alerts warning providers at the time of medication prescribing if there are potential issues with known PGx test results has great practicality. Presenting recommendations to clinicians in a timely manner, with seamless integration into clinical workflows, and easy accessibility with continually updating CDS recommendations (as the knowledge changes) is ideal. Patients often have multiple physicians, from primary care to specialists, as well as other prescribers of medications, such as dentists. Thus, ensuring some level of transportability between various EMRs needs to be explored.

As direct-to-consumer genetic testing (DTC-GT) gains popularity, the public is becoming more aware and curious about pharmacogenomics. DTC-GT is a genetic test sold directly to a consumer that can provide them with information about their genetics, including ancestry, health traits, and health risks [20]. An example of a DTC-GT is *23 and Me*, an FDA-cleared test that uses single nucleotide polymorphism genotyping to provide users with a report of health and ancestry data. This saliva test can detect SLCOB1 drug transport, CYP2C19 drug metabolism, and DPYD drug metabolism, which are genes with CPIC guidelines [21]. As of 2021, there were 26 million direct-to-consumer genetic tests performed worldwide [22].

In the United States (US), the initiative of *All of Us*, funded by the National Institutes of Health (NIH), has begun returning personalized health-related DNA results to more than 155,000 participants, with reports detailing whether participants have an increased risk for specific health conditions and how their body might process certain medications [23]. The adoption of DTC-GT and the release of data to participants from *All of Us* demonstrates the need for more robust CDST in EMR’s as the utility of pharmacogenomic tests become more pronounced and, in some instances, provided by patients directly to their providers and health systems.

This study has some limitations. The study only identified patients and prescribers from one area of Wisconsin within an academic medical setting, any regional and national prescribing trends cannot be identified. Another limitation is the study did not affirm if any PGx tests had been ordered, as this was not in the scope of the project. These limitations serve as a call for future research in this area.

## 5. Conclusions

Using existing data from an EMR is an efficient way to identify the magnitude of patients that may benefit from PGx testing as well as the disciplines of providers. Given the inherent complexity in this area (i.e., co-morbidities, multiple prescribers, various medications, and changing guidelines), critically examining results such as these enables the development of PGx testing processes, physician support strategies, patient education approaches, and pharmacist workflows to optimize the implementation of a PGx service within a medical center.

## Figures and Tables

**Figure 1 pharmacy-11-00178-f001:**
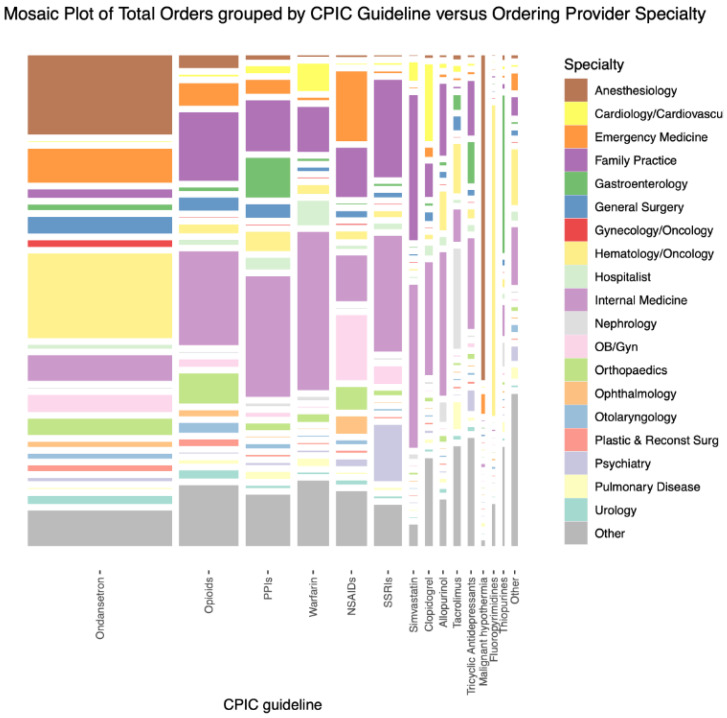
Orders of medication grouped by CPIC guideline drug compared to the ordering provider specialty.

**Table 1 pharmacy-11-00178-t001:** Demographics for patients with encounters, medication orders, and CPIC drug orders in 2015–2019.

	With Encounters			With Medication Orders			With CPIC Drug Orders		
Sex	Patients	Median Age	IQR	Patients	Median Age	IQR	Patients	Median Age	IQR
Female	463,929	47	(30–64)	327,698	46	(29–64)	190,375	50	(33–64)
Male	381,165	48	(29–64)	262,800	48	(29–64)	145,474	54	(36–64)
Other/ Unknown	424	40	(22–59)	28	34	(24–45)	BT	BT	BT

**Table 2 pharmacy-11-00178-t002:** CPIC drug ordering characteristics in 2015–2019 grouped by CPIC guidelines and patients.

			Days in Ordering Episode ^1^			Orders in Ordering Episode		
	Total Orders	Unique Patients	Median	IQR	99th Percentile	Median	IQR	99th Percentile
Ondansetron								
ondansetron	1,302,015	202,113	1	(1–57)	1245	1	(1–3)	28
Opioids								
all for guideline	508,685	137,982	NA	NA	NA	NA	NA	NA
hydrocodone	280,351	83,785	1	(1–9)	1656	1	(1–2)	32
tramadol	144,365	45,852	1	(1–41)	1615	1	(1–3)	23
codeine	83,969	43,934	1	(1–1)	1152	1	(1–2)	12
NSAIDs								
all for guideline	268,377	122,817	NA	NA	NA	NA	NA	NA
ibuprofen	186,829	95,582	1	(1–2)	956	1	(1–2)	7
meloxicam	46,007	21,696	1	(1–78)	1544	1	(1–2)	11
celecoxib	22,708	8431	1	(1–32)	1608	1	(1–2)	11
flurbiprofen	12,342	5680	1	(1–14)	385	1	(1–2)	3
piroxicam	491	172	1	(1–326)	1691	1	(1–4)	14
PPIs								
all for guideline	375,230	92,183	NA	NA	NA	NA	NA	NA
pantoprazole	246,842	66,882	2	(1–170)	1628	2	(1–4)	18
omeprazole	117,466	39,656	10	(1–500)	1704	2	(1–4)	11
lansoprazole	7557	2955	1	(1–284)	1682	1	(1–3)	11
dexlansoprazole	3365	1109	2	(1–430)	1679	2	(1–4)	12
SSRIs								
all for guideline	232,950	53,723	NA	NA	NA	NA	NA	NA
sertraline	100,702	25,105	56	(1–432)	1708	2	(1–5)	17
escitalopram	67,838	18,752	56	(1–396)	1693	2	(1–4)	16
citalopram	41,129	10,338	83	(1–602)	1727	2	(1–5)	16
paroxetine	21,904	5045	91	(1–669)	1733	3	(1–6)	18
fluvoxamine	1377	275	70	(1–429)	1764	3	(1–6)	20
Malignant hypothermia								
succinylcholine	36,058	23,895	1	(1–1)	413	1	(1–1)	4
Warfarin								
warfarin	264,046	16,036	103	(3–729)	1779	7	(3–18)	78
Simvastatin								
simvastatin	70,444	15,736	619	(1–1373)	1757	4	(2–6)	12
Clopidogrel								
clopidogrel	65,658	15,097	23	(1–416)	1701	2	(1–5)	16

^1^ One ordering episode is defined as all contiguous orders of a given drug for a patient without a gap of greater than 1.5 years.

**Table 3 pharmacy-11-00178-t003:** Combinations of CPIC drugs ordered in 2015–2019.

			Interval Days ^1^		
Total Number of CPIC Drugs	Unique Patients	Distinct Drug Combinations	Median	IQR	99th Percentile
1	133,380	47 ^2^	NA	NA	NA
2	84,334	593	4	(0–254)	1530
3	51,357	2090	172	(2–661)	1647
4	29,628	3819	457	(86–976)	1709
5	17,560	4678	694	(250–1159)	1746
6	9772	4479	875	(433–1299)	1765
7	5197	3457	1035	(617–1398)	1769
8	2600	2095	1153	(762–1473)	1779
9	1233	1133	1272	(902–1529)	1791
10+	792	776	1317	(985–1563)	1792

^1^ Ordering interval is defined as the number of days between the first order of the first drug and the earliest order of last drug in a combination for a patient. ^2^ A total of 47 distinct CPIC drugs were ordered singly for patients who received no other CPIC drugs during the study period. Note: based on the total of 335,849 patients that had medication orders for a CPIC drug.

## Data Availability

Data are contained within the article.

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
