# Peer review of "An EMR-Based Approach to Determine Frequency, Prescribing Pattern, and Characteristics of Patients Receiving Drugs with Pharmacogenomic Guidelines"

_pharmacy, 2023, doi:10.3390/pharmacy11060178_

Round 1

Reviewer 1 Report

Comments and Suggestions for Authors

Thank you for the opportunity to review "An EMR-based Approach to Determine Frequency, Prescribing 2 Pattern and Characteristics of Patients Receiving Drugs with 3 Pharmacogenomic (PGx) Guidelines (Brief Report)".

The paper reports on an important topic, and it is well-written. Although the topic is interesting, the results might not support the authors' view of urgent PGx testing.

Major:

The need for PGx testing is not clear from the paper because the paper does not study adverse effects caused by gene-drug interactions. Perhaps the authors should consider studying the actual outcomes.

From the design, it is not clear that the absence of PGx between 2015 and 2019 does not mean that the records were not there before (before 2014, as the authors seem to "look back" one year). Please describe how you ensured that PGx testing was unavailable before 2015.

It is unclear why the gap between medication orders matters and needs a careful description in Table 2.

Minor:

Figure 1 is empty.

Line  99: Possible typo. It should be "data were" instead of "data was"

Line 163: Possible typo. It should be "Hence the need" instead of "Hence the needed"

Line 164: Possible typo. It should be "will be ever" instead of "we be ever"

Line 200: Possible typo. It should be "to be explored" instead of "to be explore"

Comments on the Quality of English Language

Several typos

Author Response

Thank you for the insightful comments. Please see attachment. 

Reviewer 2 Report

Comments and Suggestions for Authors

In this manuscript, the authors investigated the EMR-based Approach to determine frequency, prescribing pattern and characteristics of patients receiving drugs with pharmacogenomic (PGx) guidelines. They concluded that identifying patient subsets that may benefit from PGx testing, will enable the development of pre-emptive testing processes, physician support strategies, and pharmacist workflows to optimize results for patients.

The authors need to address the following comments:

Abstract:

The abstract can benefit from a focused revision

Introduction

Lane 60-62: this sentence is not clear. Please rewrite clearly.

Lane 72: …done performed… please revise.

Lane 74-75: what do you mean with this sentence? once the genetic test is performed then the result is good for a lifetime and can be used by other prescribers. Please clarify.

The introduction could also benefit from a more focused revision

Results:

Figure 1. does not show the amount of medications prescribed per specialty. Please update the figure

Lane 125: clarify what do you mean by the most distinct patient?

Table 2. explain and define what do you mean by “days in ordering episode” and “orders in ordering episode”. Does it mean a number of orders per number of days? if so write it clearly

Lane 129-133. describe and revise clearly. Why is it important to look at the ordering episode and its relation to PGx.

Lane 147-143: needs clarification. Why is it important to look at the “interval days” and what is its relation to PGx and to the objective of the study which is to identify patients and prescribers who would benefit most from comprehensive PGx testing

Discussion

The whole discussion needs to revised with a focus on the thesis of the manuscript.

Comments on the Quality of English Language

The manuscript is in need of major editing and reorganization of the flow of ideas.

Author Response

(The authors gave the same response as above.)

Reviewer 3 Report

Comments and Suggestions for Authors

Figure 1 - is only description of the axis.

Round 2

Reviewer 2 Report

Comments and Suggestions for Authors

All corrections accepted

Comments on the Quality of English Language

Minor editing

Author Response

thank you for your comments.